

# The Koala (*Phascolarctos cinereus*) faecal microbiome differs with diet in a wild population

Kylie L. Brice[1,2], Pankaj Trivedi[1,2,3], Thomas C. Jeffries[1,3], Michaela D.J. Blyton[1], Christopher Mitchell[1,4], Brajesh K. Singh[1,3] and Ben D. Moore[1]

[1] Hawkesbury Institute for the Environment, Western Sydney University, Penrith, NSW, Australia
[2] Bioagricultural Sciences & Pest Management, Colorado State University, Fort Collins, CO, United States of America
[3] Global Centre for Land Based Innovation, Western Sydney University, Penrith, NSW, Australia
[4] Environment and Sustainability Institute, University of Exeter, Penryn, Cornwall, United Kingdom

Corresponding authors
Kylie L. Brice,
kylie.brice@colostate.edu,
kyliebrice72@gmail.com
Ben D. Moore,
b.moore@westernsydney.edu.au

## ABSTRACT

**Background.** The diet of the koala (*Phascolarctos cinereus*) is comprised almost exclusively of foliage from the genus *Eucalyptus* (family Myrtaceae). *Eucalyptus* produces a wide variety of potentially toxic plant secondary metabolites which have evolved as chemical defences against herbivory. The koala is classified as an obligate dietary specialist, and although dietary specialisation is rare in mammalian herbivores, it has been found elsewhere to promote a highly-conserved but low-diversity gut microbiome. The gut microbes of dietary specialists have been found sometimes to enhance tolerance of dietary PSMs, facilitating competition-free access to food. Although the koala and its gut microbes have evolved together to utilise a low nutrient, potentially toxic diet, their gut microbiome has not previously been assessed in conjunction with diet quality. Thus, linking the two may provide new insights in to the ability of the koala to extract nutrients and detoxify their potentially toxic diet.

**Method.** The 16S rRNA gene was used to characterise the composition and diversity of faecal bacterial communities from a wild koala population ($n = 32$) comprising individuals that predominately eat either one of two different food species, one the strongly preferred and relatively nutritious species *Eucalyptus viminalis*, the other comprising the less preferred and less digestible species *Eucalyptus obliqua*.

**Results.** Alpha diversity indices indicated consistently and significantly lower diversity and richness in koalas eating *E. viminalis*. Assessment of beta diversity using both weighted and unweighted UniFrac matrices indicated that diet was a strong driver of both microbial community structure, and of microbial presence/absence across the combined koala population and when assessed independently. Further, principal coordinates analysis based on both the weighted and unweighted UniFrac matrices for the combined and separated populations, also revealed a separation linked to diet. During our analysis of the OTU tables we also detected a strong association between microbial community composition and host diet. We found that the phyla Bacteroidetes and Firmicutes were co-dominant in all faecal microbiomes, with Cyanobacteria also co-dominant in some individuals; however, the *E. viminalis* diet produced communities dominated by the genera *Parabacteroides* and/or *Bacteroides*, whereas the *E. obliqua*-associated diets were dominated by unidentified genera from the family Ruminococcaceae.

**Discussion**. We show that diet differences, even those caused by differential consumption of the foliage of two species from the same plant genus, can profoundly affect the gut microbiome of a specialist folivorous mammal, even amongst individuals in the same population. We identify key microbiota associated with each diet type and predict functions within the microbial community based on 80 previously identified *Parabacteroides* and Ruminococcaceae genomes.

# INTRODUCTION

Our understanding of the contribution of the gut microbiome to digestive efficiency in vertebrate herbivores, particularly via fermentation and detoxification of low-quality diets, is rapidly improving alongside improved technology for molecular sequencing (*Flint & Bayer, 2008b*; *Kohl & Dearing, 2012*; *Miller, Kohl & Dearing, 2014*; *Suzuki, 2017*). Previous investigations into the microbiome have shown that gut microbes are essential to hosts and have identified key species that are thought to have co-evolved with their hosts and which normally maintain host homeostasis but can negatively impact the health and wellbeing of the host when disrupted through illness, antibiotic treatment or dietary dysbiosis (*Clarke et al., 2012*). To identify the role of gut microbes in host health and wellbeing, many studies create dysbiosis on a large scale through administration of high doses of antibiotics (*Kohl et al., 2014d*), substantial diet change e.g., from carnivorous to herbivorous (*David et al., 2014*), or through the addition or removal of toxic components of the diet (*Kohl et al., 2014d*). Currently, research into the gut microbiomes of specialist folivores is neglected (*Barker et al., 2013*; *Alfano et al., 2015*; *Barelli et al., 2015*).

True obligate dietary specialisation is rare in mammalian herbivores (*Shipley, Forbey & Moore, 2009*) and has been found to promote a highly-conserved but low-diversity gut microbiome in sloths (*Dill-McFarland et al., 2016*). Dietary specialists often rely on plant species that produce potentially toxic plant secondary metabolites (PSMs) which have often evolved as chemical defences against herbivory and reduce consumption by deterrence or reduction of net nutritional benefit of food (*Stupans, Jones & McKinnon, 2001*; *Moore & Foley, 2005b*; *Provenza, 2006*). However, gut microbes of specialists can sometimes also enhance tolerance of dietary PSMs, facilitating competition-free access to food sources (*Kohl et al., 2014d*). For example, *Kohl et al. (2014d)* demonstrated how microbes facilitate the intake of toxic PSMs using two populations of the specialist desert wood rat (*Neotoma lepida*), one experienced and one naïve to feeding on creosote bush (*Larrea tridentata*), which produces a resin high in the toxic PSM nordihydroguaiaretic acid. Faecal transplantation of microbes from experienced to naïve wood rats increased the tolerance of the latter group to creosote PSMs and antibiotic disruption produced a naïve gut response in experienced wood rats.

The diet of the koala (*Phascolarctos cinereus*) is comprised almost exclusively of foliage from the eucalypt genus *Eucalyptus* (family Myrtaceae). However, eucalypts produce a variety of PSMs including terpenes, cyanogenic glucosides, phenolics including condensed and hydrolysable tannins, formylated phloroglucinol compounds (FPCs) and unsubstituted B-ring flavanones (UBFs) which variously act as toxins, feeding deterrents and digestibility-reducers (*Moore et al., 2004a*; *Moore & Foley, 2005b*; *Marsh et al., 2015*; *Marsh et al., 2017*). Some PSMs, particularly monoterpenes, are potentially bactericidal (*Knezevic et al., 2016*; *Sun et al., 2018*), although the koala is a hindgut fermenter and in other hindgut-fermenting marsupials, terpenes are absorbed in the stomach and small intestine, minimising exposure of the gastrointestinal microbiome to these compounds (*Foley, Lassak & Brophy, 1987*). Concentrations and types of PSMs vary widely between eucalypt species, between individual trees and from region to region (*Moore et al., 2004b*; *Moore et al., 2004c*). Koalas living in regions with different eucalypt communities may therefore face differing nutritional and toxicological challenges (*DeGabriel et al., 2009b*). Koala microbiomes have not previously been assessed in conjunction with diet quality, although linking the two may provide new insights. The 16S rRNA gene has previously been used to describe the microbial community structure of the wild (*Barker et al., 2013*) and captive (*Alfano et al., 2015*; *Shiffman et al., 2017*) koala gut microbiome (*Barker et al., 2013*; *Alfano et al., 2015*; *Shiffman et al., 2017*), finding it to be dominated by bacteria from the phyla Bacteroidetes and Firmicutes.

## Background—Cape Otway

In southern Australia, some koala populations, usually associated with *Eucalyptus viminalis* (subgenus Symphyomyrtus), have repeatedly increased beyond the carrying capacity of their habitat, resulting in eventual population collapse and mass tree dieback (*Martin, 1985a*; *Whisson et al., 2016*). At Cape Otway, Victoria (38°50′06″S, 143°30′25″E), 75 koalas were reintroduced in 1981 and the population rapidly increased, peaking at densities of up to 18 ha$^{-1}$ in 2013, followed by a rapid population decline, or crash (*Whisson et al., 2016*). High juvenile recruitment and low mortality of koalas contribute to this phenomenon, but the high nutritional quality of *E. viminalis*, a highly-preferred food tree which dominates areas experiencing overbrowsing, is a key factor. *Eucalyptus viminalis* possesses high foliar nitrogen (N) concentrations for a eucalypt and its tannins have a negligible impact on the nutritional availability of that N (*DeGabriel et al., 2008*; *Marsh et al., 2014*), although it also possesses high FPC concentrations that can limit koala feeding (*Moore & Foley, 2005b*). At Cape Otway, a minority of koalas use the generally non-preferred *Eucalyptus obliqua* (subgenus *Eucalyptus*), usually in forest patches dominated by that species. Koalas at Cape Otway maintain very small home ranges (0.4–1.2 ha$^{-1}$; *Whisson et al., 2016*) so that small scale patchiness in tree species distributions can produce very different home range compositions. Trees from the subgenus *Eucalyptus* (''monocalypts'') do not produce FPCs (*Eschler et al., 2000*), but in contrast to subgenus *Symphyomyrtus* (''symphyomyrts''), do possess UBFs, which possibly act to deter koalas as they do another eucalypt folivore, the common brushtail possum (*Trichosurus vulpecula*, *Tucker et al., 2010*; *Marsh et al., 2015*). On average, monocalypts also possesses lower foliar available N (AvailN) concentrations, compared with symphyomyrt foliage, due to greater protein precipitation by tannins

(*Wallis, Nicolle & Foley, 2010*). These patterns present an opportunity to compare and contrast the gut microbial communities of koalas consuming two different eucalypt diets.

Here, we aim to: (a) determine the impact of two eucalypt (*E. viminalis* and *E. obliqua*) diets on the relative abundance of dominant bacterial groups within the koala gut and; (b) assess the diet quality of these two species. We hypothesised that the gut microbiomes of koalas eating different eucalypt diets would differ in community composition. To test our hypothesis, we collected faecal material (pellets) from a single koala population that included individuals believed to consume two different eucalypt species, during the population peak in 2013 ($n = 14$), and again, from different individuals, post-population collapse in 2015 ($n = 18$). We characterised the faecal microbial community as a proxy for gut microbial community composition (*Amato et al., 2013*; *David et al., 2014*) using both terminal restriction fragment length polymorphism, and 16S rRNA amplicon sequencing. Diet composition was estimated by faecal plant wax analysis.

## EXPERIMENTAL METHODS

Detailed experimental procedures can be found in the online version of this article under Supplementary Information.

### Study species

The koala is a specialist folivorous marsupial that is digestively and morphologically adapted to its challenging diet of eucalypt foliage. Koalas have an extended caecum and proximal colon which function as the major site of microbial fermentation (*Cork & Warner, 1983a*; *Snipes, Snipes & Carrick, 1993*). In addition, they selectively retain solutes and small particulate matter in their hindgut for extended periods of up to 213 h (*Krockenberger & Hume, 2007*). These adaptations increase the exposure of digesta to microbial fermentation, while reducing the amount of microbial protein lost in faecal matter (*Foley & Cork, 1992*; *Hume & Esson, 1993*; *Hume, 2005*). Koala faecal material is expelled as hard, dry pellets.

### Faecal sample collection

Faecal pellets were collected from single a contiguous koala population occupying a four-square kilometre area of the Cape Otway (38°50′06″S, 143°30′25″E) Peninsula of Victoria, Australia in two collection years, the first from koalas ($n = 14$) during February and September 2013 (see Fig. S1), and the second from koalas ($n = 18$) during January 2015 (see Fig. S2). The peak of the koala population boom occurred at Cape Otway in 2013 and resulted in the eventual death by starvation of hundreds or thousands of koalas (*Whisson et al., 2016*). In 2013, most koalas in *E. viminalis* patches had limited or no access to adult foliage and subsisted on epicormic regrowth leaves. These individuals may have experienced some degree of malnourishment. However, by 2015, koala population density had declined, and samples were collected from areas with healthier *E. viminalis* canopies containing a mixture of adult and epicormic foliage.

Faecal samples were collected from koalas ($n = 32$) inhabiting forest patches locally dominated by or exclusively containing either *E. viminalis* ($n = 19$) or *E. obliqua* ($n = 13$; see Figs. S1 & S2), and in most cases koalas were observed feeding in the trees they were

occupying. Koalas in *E. viminalis* forest at Cape Otway occupy very small (0.4–1.2 ha$^{-1}$, *Whisson et al., 2016*) home ranges, allowing us to be confident that *E. viminalis* was the dominant or only food source available for those areas, and similarly in *E. obliqua* areas, *E. viminalis* was absent or killed. In 2013, koalas were present in high densities and signs of overbrowsing or severe defoliation were universal in *E. viminalis* patches while koalas were much less abundant and overbrowsing was less apparent in *E. obliqua* patches.

Mats were placed under koalas in trees and checked at intervals of no more than four hours throughout the day. Fresh faecal pellets were counted and collected from mats, placed into zip-lock bags and placed on ice until they could be transferred, within 2 h, into a −20 °C freezer for storage. Studies into the impact of storage conditions on results of 16S ribosomal RNA (rRNA) sequencing from faecal material have concluded that phylogenetic structure and community diversity are not significantly impacted by either short-term storage at 4 °C or 20 °C (<24 h), or long-term storage at −20 °C or −80 °C (*Carroll et al., 2012*; *Lauber et al, 2010*).

Due to the non-invasive nature of faecal collection, faeces have been extensively used to study gut microbial composition and ecology in humans and animals during both health and disease (*Turnbaugh et al., 2009a*; *David et al., 2014*; *Dill-McFarland et al., 2016*; *Lichtman, Sonnenburg & Elias, 2015*). Previous studies have found differences in microbial community composition between gastrointestinal and faecal samples (*Stearns et al., 2011*; *Alfano et al., 2015*). The use of faeces as a proxy for the gut microbial community has nonetheless revealed important and biologically meaningful findings in both humans and animals (*Amato et al., 2013*; *David et al., 2014*) and is the only non-invasive method available for sample collection from wild animals.

As the field work described in this manuscript did not require either the capture or disturbance of koalas, no animal ethics application or scientific research permit was required. All fieldwork was conducted on private land, meaning that no research permit was required under Victorian National Parks legislation.

## Analysis of bacterial community through rRNA gene sequencing and analysis

Amplification and sequencing of the V4 region of the bacterial 16S rRNA gene was undertaken using a previously established protocol and primers 515F and 806R from *Caporaso et al. (2011a)*. Paired-end 16S rRNA community sequencing was performed using the Illumina MiSeq® platform at the Ramaciotti Centre for Genomics (UNSW, Australia; 2013 koala faecal samples) and the Next-Generation Sequencing Facility at Western Sydney University (Richmond, Australia; 2015 koala faecal samples), using the same protocol, primers and Illumina platform. Analyses of sequence data were performed using the Quantitative Insights into Microbial Ecology (QIIME) pipeline, version 1.8 (*Caporaso et al., 2010b*). Sequences were quality-checked and low quality (<Q30) sequences were removed from further analysis. Sequences were aligned against the Greengenes 13_8-release database (*DeSantis et al., 2006*) and potentially chimeric sequences (∼4% of total sequences) were removed using Chimera Slayer (*Haas et al., 2011*). Sequences were aligned (PyNAST, *Caporaso et al., 2010a*), and clustered (uclust, *Edgar, 2010*) into operational

taxonomic units (OTUs) defined as sharing 97% sequence identity (hereafter, 'taxa'). Samples were rarefied to the smallest dataset consisting of 107,813 sequences, and alpha diversity measures including Chao1 (measure of species richness), and Shannon indexes (diversity) were calculated (*Good, 1953*).

Beta diversity and relative abundances of taxa were assessed using the phylogenetic distance-based measurement weighted and unweighted UniFrac (diversity, *Lozupone et al., 2011*). Taxonomic identities at all levels were assigned by default in QIIME using the Ribosomal Database Project (RDP; *Wang et al., 2007*). PRIMER v 7.0.13 (*Clarke, 1993*) and PERMANOVA+ (*Anderson, 2001*) were used to conduct multivariate statistical analysis of the summarised OTU tables generated through QIIME as described for T-RFLP analysis (see supplementary information for details). Multivariate statistical analysis of the weighted and unweighted UniFrac matrices were also conducted using PRIMER v 7.0.13 (*Clarke, 1993*) and PERMANOVA+ (*Anderson, 2001*) without data transformation. PERMANOVA models used for analysis of the influence of diet on combined beta diversity included diet: fixed and collection: random, and for analysis of the influence of diet on OTU tables from individual collection years included diet: fixed and koala: random. PCoAs were performed on the UniFrac matrices in R and plotted using ggplots (*Wickham, 2009*; *R Development Core Team , 2013*). The variation in total microbial relative abundance across the samples was assessed using ANOVA and Tukey's post-hoc tests.

## Cyanobacteria identity and chloroplast contamination assessment

To confirm identity of sequences identified as Cyanobacteria, we used the CLC genomics workbench software v 7.5 (CLC bio) to perform a basic local alignment search-nucleotide (BLASTn; *Altschul et al., 1990*) analysis on the 16S rRNA gene sequences, and the previously identified rumen bacterium *YS2* 16S rRNA gene (accession number AF544207). To check for chloroplast contamination, a BLASTn analysis was performed on the 16S rRNA gene sequences, identified as Cyanobacteria *YS2*, and the 16S rRNA gene sequence from *Eucalyptus grandis* chloroplasts (accession number HM347959.1, *Paiva et al., 2011*).

## Leaf collection method

Eucalyptus respond to severe defoliation by producing abundant epicormic growth, which is ontogenetically more similar to juvenile than to typical adult foliage. In the case of *E. viminalis*, epicormic foliage accounted for the majority of foliage available to koalas in 2013. To assess the relative nutritional composition of the two-eucalypt species at Cape Otway, we collected epicormic and adult *E. viminalis* ($n = 16$) and *E. obliqua* ($n = 11$) leaves from trees koalas were located in during September 2013. Leaves were placed into labelled zip-lock bags and frozen at $-20\,°C$ until processing. They were subsequently freeze-dried and ground to pass a 1 mm screen using a CT 193 Cyclotec$^{TM}$ Sample Mill (Foss, Mulgrave, Victoria, Australia).

## Diet composition

The chemical composition of cuticular wax often differs among the foliage of different species of higher plants. Due to their ability to traverse the gastrointestinal tract relatively intact, *n*-alkanes are the most frequently used wax marker in diet composition studies

(*Dove & Mayes, 2005*). Therefore, we used the *n*-alkane protocol described by *Dove & Mayes (2005)* to estimate diet composition. Analysis was performed on an Agilent 7890A gas chromatograph coupled with an Agilent 5975C MSD (Agilent Technologies Pty Ltd, Mulgrave, VIC, Australia; details provided in the supplementary information). To estimate koala diet composition, 6 *n*- alkane peaks ($C_{23}$, $C_{25}$, $C_{27}$, $C_{28}$, $C_{29}$ and $C_{31}$) were identified and quantified in leaf ($n = 4$ samples per species) and faecal samples using the Agilent MSD Chemstation v E.02.02 software package (Agilent Technologies Pty Ltd). Estimates of diet composition were determined following the calculations protocol described by *Dove & Mayes (2005)*.

## Diet quality

Nitrogen, tannins and fibre are key determinants of herbivore diet quality. In the common brushtail possum *(Trichosurus vulpecula)*, a generalist marsupial folivore that also eats *Eucalyptus*, a measure of dietary available N (AvailN, *DeGabriel et al., 2009b*) determined using an *in vitro* digestion assay, has successfully predicted reproductive success of free-living individuals (*DeGabriel et al., 2009a*). We implemented the assay described by *DeGabriel et al. (2008)*, which returns the following measures of foliar quality: in vitro dry matter digestibility (DMD), total nitrogen concentration (N) and available, or digestible, N (AvailN). We analysed both epicormic (*E. viminalis* $n = 9$ and *E. obliqua* $n = 5$) and adult (*E. viminalis* $n = 7$ and *E. obliqua* $n = 6$) leaf samples. Total N concentrations for leaf material and their residues after digestion, were measured by a combustion method based on the Dumas method (*Wright & Bailey, 2001*) using the Leco C/N analyser (Leco Tru Mac® Corporation, Michigan, USA). Analysis of the in vitro digestion data was conducted using PRIMER v 7.0.13 and PERMANOVA+ to assess overall differences in quality between the two diets, for assessment of the differences within and between diet variables including mean total N, mean DMD, mean N digestibility (Ndig) and mean AvailN, the Students *t*-test was used.

## Comparison of potential functional differences between *Parabacteroides* and Ruminococcaceae genomes

The two dominant groups of bacteria displaying the largest change in relative abundance between koala diet groups were the genus *Parabacteroides* and family Ruminococcaceae (see 'Results'). To give an indication of the potential functional changes in the microbial community, we accessed complete or draft linear genomes (at least 30× coverage) of 35 bacteria from *Parabacteroides* and 45 from Ruminococcaceae (see Table S5) with the "Genome Browser" from the Microbial Genome and Metagenome Data Analysis pipeline of the Department of Energy Joint Genome Institute (DOE JGI) site (https://img.jgi.doe.gov/cgi-bin/m/main.cgi). We assessed the relationship between the 16S rRNA gene sequences, from the previously identified *Parabacteroides* and Ruminococcaceae genomes and those isolated from the koalas' microbiomes. We found that the koala 16S rRNA gene sequences were dispersed throughout the constructed *Parabacteroides* and Ruminococcaceae phylogenetic trees (see Figs. S8 & S9). The genomes were analysed for the presence of genes encoding enzymes and transporters and grouped to glycoside

hydrolase (GH) families according to the substrate specificities of characterised enzymes, as classified in the carbohydrate-active enzymes (CAZy) database (*Cantarel et al., 2008*; *Berlemont & Martiny, 2015*). GH families were classified as those targeting oligosaccharides, starch and glycogen, cellulose, xylan, chitin, dextran, fructan, or other animal or plant polysaccharides; and structural polysaccharides (i.e., cellulose, chitin, and xylan) as described by *Berlemont & Martiny (2015)*. Next, genomes were classified according to their potential for oligosaccharide and polysaccharide processing. Potential degraders of these substrates were defined as bacteria having at least one gene targeting one of these specific substrates.

## RESULTS

### Bacterial community rRNA gene sequencing

A total of 33,102,252 reads with 31,832,423 remaining post-sequence quality control were obtained from the 32 samples sequenced (see Tables S1 and Tables S2A). Faecal microbial communities of koalas eating *E. viminalis* were significantly less diverse (Shannon $4.60 \pm 0.41$ vs $5.30 \pm 0.25$; ANOVA $P = 0.001$) and less taxonomically rich (Chao $8,100 \pm 2,652$ vs $10,313 \pm 2079$; ANOVA $P = 0.001$) than those of koalas eating *E. obliqua* (see Table S2B). Evenness did not change between collection years, but richness in 2015 was higher than in 2013 (Chao $P = 8.68e-13$; Shannon $P = 0.121$). We assessed whether differences between the faecal bacterial communities associated with the two diets were driven by community structure, i.e., differences in relative abundance of taxa (weighted UniFrac, *Lozupone et al., 2011*), or by altered presence/absence of microbial taxa (unweighted UniFrac) using the 16S rRNA genes retained after removal of chimeric sequences. These β-diversity matrices assess the extent of branch sharing on a master phylogenetic tree, weighting branches by the relative abundance of taxa (*Lozupone et al., 2011*). The UniFrac PCA biplots revealed a clear separation based on the PC1 and PC2 axes of the weighted scatter plot, explaining 63% of the total variation between the two combined collections (Fig. 1A), reinforcing the observations from T-RFLP analysis (see Fig. S3). The first two components of the unweighted scatter explained 18% of the total variation (Fig. 1B). PERMANOVA analysis indicated no influence of collection year on the weighted UniFrac data (*pseudo* $F_1 = 0.44$, PERMANOVA $P = 0.81$), but a significant influence was detected when we analysed the unweighted UniFrac data (*pseudo* $F_1 = 1.76$, PERMANOVA $P = 0.01$). When we analysed diet × collection year an influence was detected on community structure (weighted UniFrac, *pseudo* $F_2 = 10.07$, PERMANOVA $P = 0.0001$) and presence/absence of microbial taxa (unweighted UniFrac, *pseudo* $F_2 = 2.98$, PERMANOVA $P = 0.0001$).

PERMANOVA assessment of the weighted and unweighted UniFrac matrices from individual collection years (2013 and 2015), indicated that diet was a strong driver of both microbial community structure (relative abundance), and of microbial presence/absence during 2013 (weighted UniFrac, *pseudo* $F_1 = 5.88$, PERMANOVA $P = 0.0001$; unweighted UniFrac, *pseudo* $F_1 = 1.89$, PERMANOVA $P = 0.0001$). The influence of diet was also significant in 2015 (weighted UniFrac, *pseudo* $F_1 = 8.89$, PERMANOVA $P = 0.0001$;

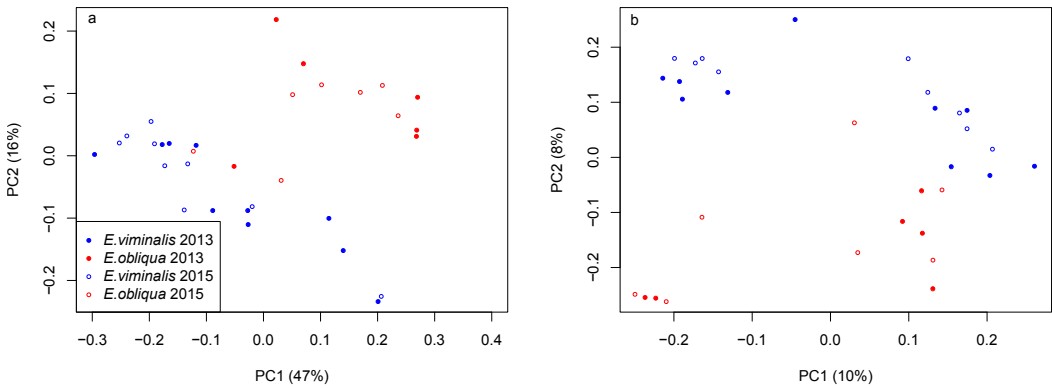

**Figure 1** **PCoA of β diversity of koala faecal bacterial communities from Cape Otway.** Scatterplots from (A) weighted and (B) unweighted UniFrac matrices for the combined Cape Otway koala population with diets comprising *E. viminalis* and *E. obliqua* from 2013 and 2015. PERMANOVA analysis indicated no influence of collection year on the (A) weighted UniFrac data (*pseudo* $F_1 = 0.44$, PERMANOVA $P = 0.81$). A significant influence was detected when we analysed the (B) unweighted UniFrac data (*pseudo* $F_1 = 1.76$, PERMANOVA $P = 0.01$). When diet × collection years was analysed an influence was detected on community structure ((A) weighted UniFrac, *pseudo* $F_2 = 10.07$, PERMANOVA $P = 0.0001$) and presence/absence of microbial taxa ((B) unweighted UniFrac, *pseudo* $F_2 = 2.98$, PERMANOVA $P = 0.0001$).

unweighted UniFrac, *pseudo* $F_1 = 2.76$, PERMANOVA $P = 0.0001$). PCoA analysis of the weighted and unweighted UniFrac matrices for the 2013 koala collection, showed a clear separation linked to diet, with 75% of total variation explained by PC1 (weighted UniFrac, Fig. 2A), while the unweighted UniFrac matrix also indicated an influence of diet, although the two axes explained less (30%) of the total variation (Fig. 2B). Overall the separation between the microbiomes of the *E. viminalis* and *E. obliqua* koalas was smaller in 2015 than 2013 (average pairwise Unifac distance between groups ± standard deviation, weighted: 2013 = 0.481 ± 0.082; 2015 = 0.377 ± 0.075; unweighted: 2013 = 0.756 ± 0.072; 2015 = 0.666 ± 0.073). This was reflected in the 2015 PCoA analysis (Figs. 3A & 3B). PC1 and PC2 of the weighted and unweighted PCoAs explained 65 and 18% of the total variation, respectively. Similar patterns were observed during T-RFLP analysis of the two collection years (see Figs. S4A & S4B).

Analysis of phylum-level OTUs indicated a separation between diets for both 2013 and 2015 collections (2013, *pseudo* $F_1 = 65.47$, PERMANOVA $P = 0.0001$; 2015 *pseudo* $F_1 = 24.95$, PERMANOVA $P < 0.0001$ respectively). In 2013, there was an almost three-fold increase in relative abundance of Bacteroidetes in *E. viminalis* koalas, compared with an almost three-fold increase in relative abundance of Firmicutes in the *E. obliqua* koalas (Table 1). The same pattern in relative abundance was observed in the 2015 koala faecal microbiomes, although to a lesser extent (Table 1). There were also significant differences in the relative abundance of some phyla between collection years. For example, the phylum Synergistetes was more abundant in the faecal microbiomes of koalas eating both *E. viminalis* and *E. obliqua* in 2015 (Table 1). Cyanobacteria increased in relative abundance in the 2015 *E. viminalis* faecal microbiomes by 15× that of the *E. viminalis* 2013 faecal microbiomes, where *E. obliqua* faecal microbiomes had an 75% increase in

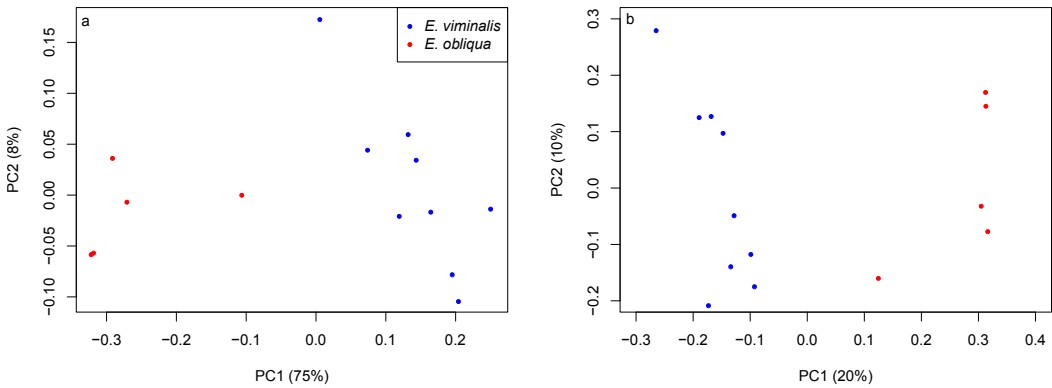

**Figure 2** **PCoA of β diversity based on UniFrac matrices from the Cape Otway 2013 koala collection.** Scatterplots from (A) weighted and unweighted (B) UniFrac matrices for koalas with diets of *E. viminalis* and *E. obliqua* from the 2013 collection year. PERMANOVA assessment of the weighted and unweighted UniFrac matrices from the 2013 collection year, indicated that diet was a strong driver of both microbial community structure (relative abundance), and of microbial presence/absence during 2013 (A) weighted UniFrac, *pseudo* $F_1 = 5.88$, PERMANOVA $P = 0.0001$; (B) unweighted UniFrac, *pseudo* $F_1 = 1.89$, PERMANOVA $P = 0.0001$.

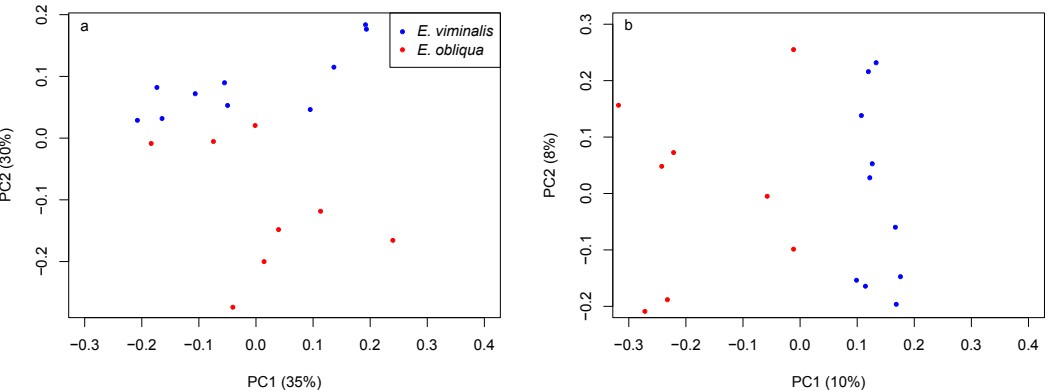

**Figure 3** **PCoA of β diversity based on UniFrac matrices from the Cape Otway 2015 koala collection.** Scatterplots from (A) weighted and (B) unweighted UniFrac matrices for koalas with diets comprising *E. viminalis* and *E. obliqua* from the 2015 collection year. The influence of diet was significant in 2015 (A) weighted UniFrac, *pseudo* $F_1 = 8.89$, PERMANOVA $P = 0.0001$; (B) unweighted UniFrac, *pseudo* $F_1 = 2.76$, PERMANOVA $P = 0.0001$.

relative abundance of Cyanobacteria (Table 1). Interestingly, the difference in relative abundance of Cyanobacteria between the two diets reversed in 2015 (Table 1). BLASTn analysis of the sequences identified as Cyanobacteria order *YS2* confirmed that they are Cyanobacterial sequences (between 91 and 96% identity with the 16S rRNA gene from rumen bacterium *YS2* accession number AF544207) and not chloroplast contamination (between 74 and 88% identity with the 16S rRNA gene from *Eucalyptus grandis* chloroplast genome accession number HM347959.1 (*Paiva et al., 2011*).

**Table 1** Relative abundance of several bacterial phyla in faeces of koalas eating *E. viminalis* and *E. obliqua* in 2013 and 2015.

| | Frequency (%) ± SE | | | |
|---|---|---|---|---|
| Taxon-phylum | *E. viminalis* 2013 | *E. obliqua* 2013 | *E. viminalis* 2015 | *E. obliqua* 2015 |
| Bacteroidetes | 61 ± 0a | 22 ± 0b | 48 ± 0a | 27 ± 0b |
| Firmicutes | 34 ± 0b | 69 ± 0a | 32 ± 0b | 61 ± 0a |
| Cyanobacteria | 1 ± 0b | 6 ± 0ab | 15 ± 0a | 8 ± 0ab |
| Other | 1 ± 0b | 1 ± 0a | 0.003 ± 0.001c | 0.003 ± 0.001c |
| Proteobacteria | 1 ± 0a | 1 ± 0a | 1 ± 0a | 1 ± 0a |
| Synergistetes | 0.003 ± 0.001b | 0.0004 ± 0.0001b | 2 ± 0a | 1 ± 0b |
| Verrucomicrobia | 2 ± 0a | 0.00004 ± 0.00002a | 1 ± 0a | 1 ± 0a |

**Notes.**
Values given as mean relative abundance ± standard error (SE) across diet and collection year. Across columns, mean followed by the same letter are not significantly different at $P < 0.05$, using Tukey's test.

Analysis of genus-level OTUs also revealed a significant separation between koala diets in 2013 (*pseudo* $F_1$ = 56.08, PERMANOVA $P$ = 0.001; see Fig. S5A) and 2015 (*pseudo* $F_1$ = 17.04, PERMANOVA $P$ = 0.001; see Fig. S5B). In 2013, there were notable diet-associated differences in relative microbial abundance, in particular, the *E. viminalis* koala faecal microbiomes were dominated by *Parabacteroides* (52 ± 4; Fig. 4A; see, Table S3), while the faecal microbiomes of *E. obliqua* koalas were dominated by an unknown genus from the family Ruminococcaceae (45 ± 1; Fig. 4A; see, Table S3). BLASTn database searches (*Altschul et al., 1990*) were unable to improve taxonomic resolution.

In 2015, the genus dominating faecal microbiomes of koalas eating *E. viminalis* was *Bacteroides* (25 ± 5; Fig. 4B; see, Table S3). The relative abundance was 3.5× that seen in the 2013 *E. viminalis* koala faecal microbiomes, while the relative abundance of *Parabacteroides* was 42% lower in the 2015 *E. viminalis* koala faecal microbiomes compared with 2013 (Figs. 4A & 4B; see, Table S3). In both 2013 and 2015, *E. obliqua* faecal microbiome communities were dominated by the family Ruminococcaceae (31 ± 4; Fig. 3A & 3B; see Table S4). However, the 2015 *E. obliqua* faecal bacterial communities had almost four × the *Parabacteroides* (19 ± 4), and half the *Bacteroides* relative abundances compared with 2013 *E. obliqua* faecal microbial communities (Figs. 4A & 4B; see, Table S3). Other genera represented in both 2013 and 2015 faecal microbiomes at relative abundances from 3 to 13% include *Acidaminococcus*, *Akkermansia*, *Coprobacillus*, *Clostridium*, *Oscillospira* and *Ruminococcus*, (Figs. 4A & 4B; see, Table S3). Relative abundance of these OTUs showed differences within and/or between the two collection years and diet types, while the remaining genera did not show significant differences (Figs. 4A & 4B; see Table S3).

## Functional differences between previously identified and publicly available *Parabacteroides* and Ruminococcaceae genomes

Due to the dominance and significant changes in relative abundance of *Parabacteroides* and Ruminococcaceae we analysed 35 previously identified and publicly available, *Parabacteroides* and 45 Ruminococcaceae genomes and identified glycoside hydrolase (GH) families involved in the degradation of plant cell walls, starch and other components ranging from easily degraded to recalcitrant (see Fig. S7). In general, we found that the

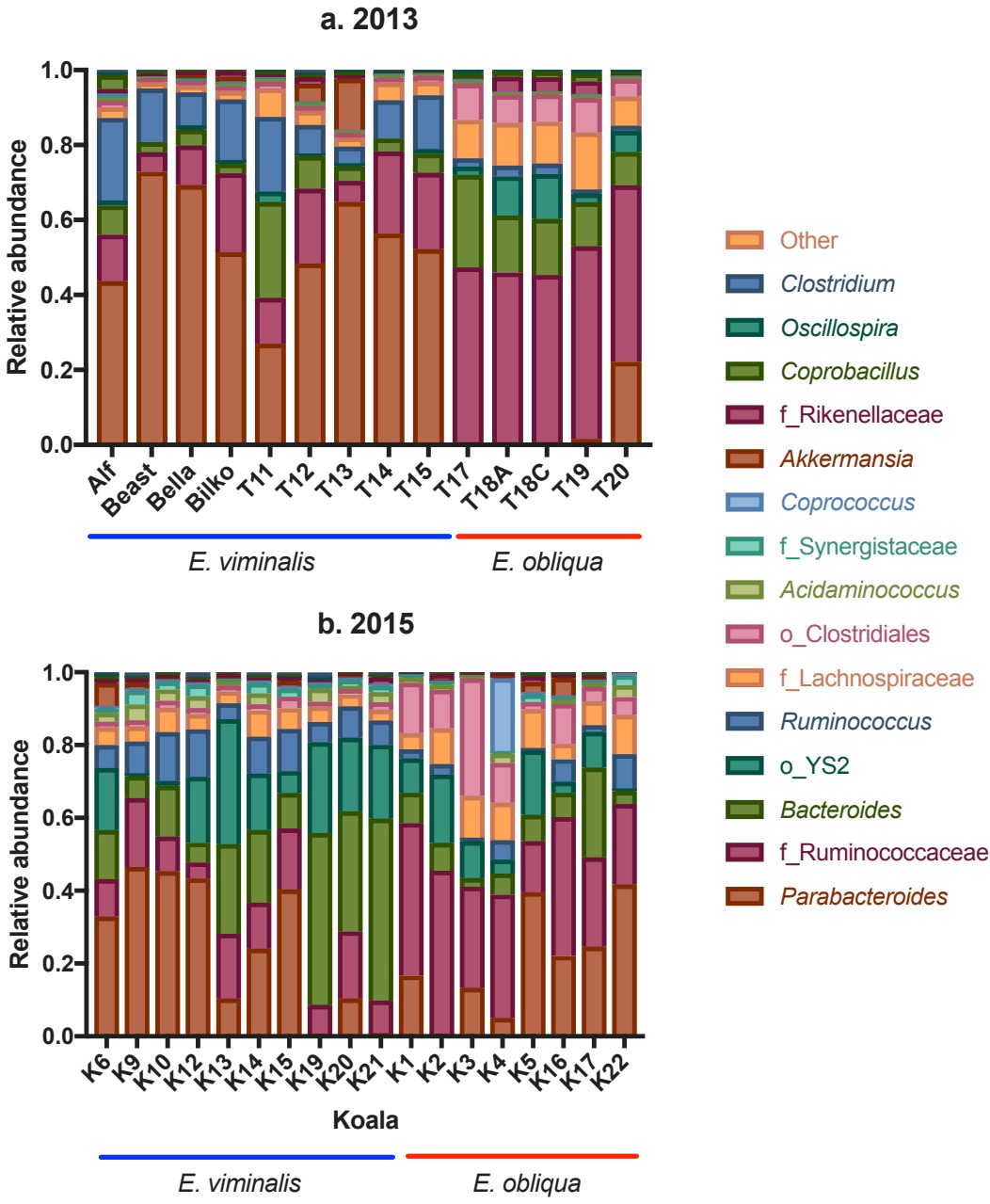

**Figure 4** **Taxonomic bar charts of the most abundant genera in faecal microbiomes of koalas feeding from *E. viminalis* and *E. obliqua*.** Taxonomic bar charts show the dominance of *Parabacteroides* and Ruminococcaceae in faecal microbiomes of koalas eating *E. viminalis* and *E. obliqua*, respectively, in (A) 2013, and the dominance of *Bacteroides*, *Parabacteroides* (*E. viminalis*) and Ruminococcaceae (*E. obliqua*) in (B) 2015.

*Parabacteroides* genomes (associated with *E. viminalis* diets) possessed more genes for oligosaccharide degradation than Ruminococcaceae genomes, while Ruminococcaceae genomes (more strongly associated with *E. obliqua* diets) possessed up to five × the number of enzymatic genes targeting the degradation of recalcitrant cellulose (see Fig. S7).

*Parabacteroides* genomes also have more genes from the GH67 and GH85 families, which are involved in the degradation of xylan and chitin, respectively. Interestingly, *Parabacteroides* genomes potentially have more genes associated with tannin degradation than Ruminococcaceae (see Fig. S7). ATP-binding cassette (ABC) and phosphotransferase system (PTS) transporter genes were more common in Ruminococcaceae (average of 134 ABC and 18 PTS per genome, see, Table S5) than in *Parabacteroides* (82 ABC and 6 PTS, see, Table S5).

## Diet composition

Because of the strong association observed between diet and microbiome, we analysed koala faecal pellets to confirm our expectation of diet composition (based upon koala location and tree occupancy), using methods established to estimate composition of diets using the *n*-alkanes that occur in the leaf cuticle of all plants as markers (*Dove & Mayes, 2005*).

Analysis of *n*-alkane markers in *E. viminalis* and *E. obliqua* leaf material showed significantly different profiles between species, with *E. obliqua* cuticle dominated by the $C_{27}$ chain-length *n*– alkane ($78 \pm 3\%$ of total alkanes) and *E. viminalis* dominated by $C_{29}$ ($91 \pm 3\%$). The relative abundance of the $C_{25}$ alkane was also 7 times greater in *E. obliqua* than *E. viminalis*. Koala faecal alkane profiles clearly separated our two diet categories and largely confirmed our expectations about diet composition based on tree canopy composition at koala pellet collection sites. In 2013, the *n*-alkane method estimated that 8/9 "*E. viminalis*" koalas included >80% *E. viminalis* in their diets, and 5/5 "*E. obliqua*" koalas included >80% *E. obliqua* (Fig. 5A). In 2015, 8/8 "*E. viminalis*" diets were estimated to contain >70% *E. viminalis*, and 6/7 "*E. obliqua*" diets >50% *E. obliqua* (Fig. 5B).

## Diet quality

Nutritional analysis of *E. viminalis* and *E. obliqua* epicormic and adult leaves indicated that overall *E. viminalis* provided koalas with foliage of higher nutritional quality (i.e., greater *in vitro* dry matter digestibility (DMD), available (or digestible) N (AvailN) and total foliar N concentrations; *pseudo* $F_1 = 36.43$ and PERMANOVA $P = 0.001$) compared with *E. obliqua* foliage (Table 2). The lower AvailN of *E. obliqua* is due to both lower total N concentrations and lower N digestibility (NDig), the latter indicating a stronger effect of anti-nutritional tannins (*DeGabriel et al., 2008*). We confirmed that adult *E. viminalis* foliage contained less recalcitrant fibre (i.e., hemicellulose, lignin and cellulose) than adult *E. obliqua* foliage, using the neutral detergent fibre (NDF) assay (*Van Soest, Robertson & Lewis, 1991*; $P = 0.003$, Table 2).

## DISCUSSION

Dramatic diet changes in humans e.g., switching from carnivorous to herbivorous diets, can profoundly affect the microbial community structure in the gut (*Ley et al., 2008a*; *Ley et al., 2008b*; *David et al., 2014*). Here we show that equally dramatic effects can be produced, even within a continuous animal population, through a difference in consumption between one and another congeneric tree species. Bacterial communities associated with each of the two diets differed primarily in the relative abundance of two phyla, Bacteroidetes and

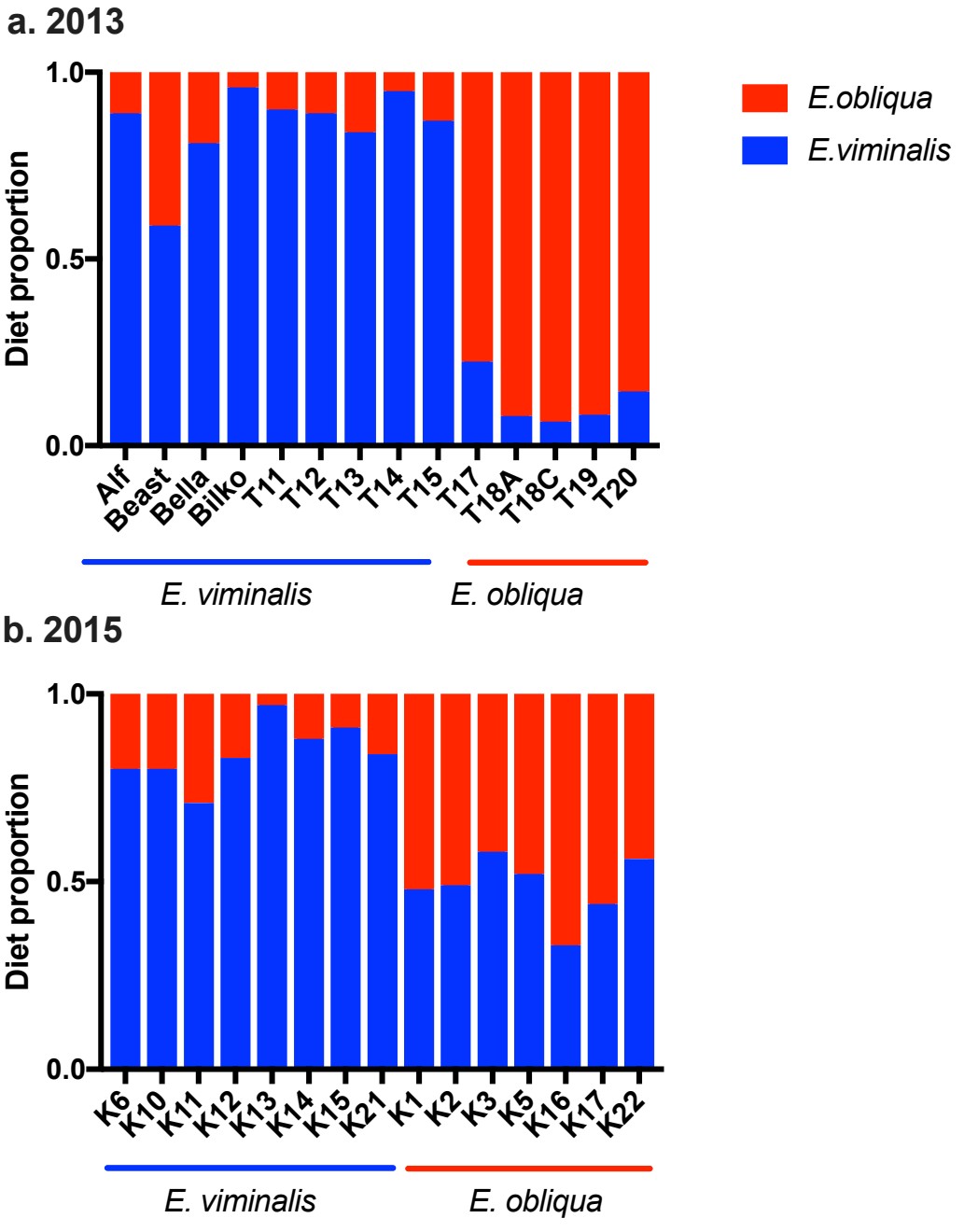

**Figure 5** **Estimated diet composition of koalas eating *E. viminalis* and *E. obliqua*.** Bar chart showing koala identity and percentage of diet species eaten for (A) 2013 ($n = 14$ koalas) and (B) 2015 ($n = 15$ koalas) koala faecal samples.

Firmicutes, rather than in presence and absence of abundant taxa similar to changes seen in humans when undergoing major diet change (*David et al., 2014*). Alpha diversity was significantly lower in terms of Shannon diversity and richness (Chao1) in koalas eating the nutritionally superior *E. viminalis*. Low microbial richness has been convincingly linked to

**Table 2  Foliar nutritional quality of leaf material analysed through *in vitro* digestion.**

| Leaf | Mean foliar nutritional values % ± SE | | | | | |
| --- | --- | --- | --- | --- | --- | --- |
| | *E. viminalis* epicormic | *E. obliqua* epicormic | P | *E. viminalis* adult | *E. obliqua* adult | P |
| Total N | 2.03 ± 18 | 1.3 ± 0.4 | 0.04 | 2 ± 0.02 | 1.19 ± 3 | 0.05 |
| Ndig | 62 ± 11 | 6.3 ± 2.6 | 0.0001 | 66 ± 3 | 15 ± 5 | 0.0001 |
| AvailN | 1.30 ± 33 | 1 ± 0 | 0.001 | 1.14 ± 0.16 | 0.17 ± 0.06 | 0.001 |
| DMD | 54 ± 3 | 31 ± 2 | 0.0001 | 52 ± 0 | 35 ± 2 | 0.0001 |
| NDF | – | – | – | 38 ± 2 | 48 ± 1 | 0.003 |

**Notes.**
Ndig (N digestibility), N (total nitrogen), Avail N (Available nitrogen), DMD (dry matter digestibility) and NDF (neutral detergent fibre). *E. viminalis* ($n = 16$) and *E. obliqua* ($n = 11$). P values obtained by Student's *t* test assessing differences between same leaf ontogeny (i.e., epicormic or adult) of both eucalypts.

higher feed efficiency in ruminants (*Shabat et al., 2016*). In our study, lower diversity and richness is likely to be associated with a greater energy harvest from *E. viminalis* compared to *E. obliqua*. Efficient microbiomes are often less complex but more specialised, which creates higher availability of ecosystem goods, such as energy resources, to the host (*De Groot, Wilson & Boumans, 2002*; *Shabat et al., 2016*). In addition, the general rearrangement of taxa within the less diverse phylum Bacteroidetes and the more diverse Firmicutes (*Parks et al., 2018*) in response to change in diet, could also contribute to lower richness and diversity in koalas eating *E. viminalis* compared to *E. obliqua*. Southern Australian koalas from *E. viminalis* forests and woodlands are unusual in specialising on a single eucalypt species, but such specialisation over many generations may have resulted in an equally unusual, specialised microbiome.

Associations between diet and the relative abundance of Bacteroidetes and Firmicutes have been linked to the functional potential of the gut microbiome in humans and animals (*Muegge et al., 2011*). For instance, Bacteroidetes genomes encode GH enzymes targeting a wide variety of relatively easily degraded plant components including non-cellulosic polysaccharides, oligosaccharides and glycogen (*Hooper, Midtvedt & Gordon, 2002*). In contrast, Firmicutes genomes encode GHs targeting cellulases and xylanases that would be beneficial in an environment dominated by recalcitrant fibre (*Ben David et al., 2015*). High prevalence of Firmicutes and a diminished relative abundance of Bacteroidetes has been associated with adjustments of the microbiome to increased fibre intake and reduced protein consumption during transitions from animal to plant diets (*David et al., 2014*; *Ben David et al., 2015*). Similarly, shifts in the Bacteroidetes: Firmicutes ratio in koalas eating *E. obliqua* could be due to the lower available protein and higher fibre content of *E. obliqua* as compared to *E. viminalis*.

We suggest several further potential functional differences in microbiomes associated with diet in koalas. Koalas eating *E. viminalis* hosted bacterial communities dominated by *Parabacteroides*, the genomes of which usually encode multiple enzymes that sense, bind and metabolise a variety of oligosaccharides (*Mahowald, 2010*). These may represent a larger fraction of leaf material in *E. viminalis* foliage, which is 50% more digestible *in-vitro* than *E. obliqua,* which has more recalcitrant cell wall components compared with adult

leaf foliage from *E. viminalis* (48 and 38% respectively). However, caution is required in interpretation of the NDF results as tannins are known to inflate NDF measurements, leading to an overestimation of fibre (*Makkar et al., 1995*). The microbiome associated with *E. obliqua* was dominated by the family Ruminococcaceae. Ruminococcaceae have smaller genomes than *Parabacteroides*, with fewer glycan-degrading enzymes, and are suited to the degradation of more varied dietary carbohydrates (*Biddle et al., 2013*; *Ben David et al., 2015*). They have more ABC and PTS transporters that may provide a competitive advantage over *Parabacteroides*, by facilitating faster bacterial uptake of sugars (*Biddle et al., 2013*). The gut lumen of koalas eating *E. obliqua* would have higher concentrations of tannins and recalcitrant fibre, so while the observation of greater cellulose-degrading functionality associated with Ruminococcaceae is expected, the potentially greater tannin-degrading functionality of *Parabacteroides* is more surprising. Little is known of qualitative variation in tannin composition among eucalypt species.

Cyanobacteria (the order *YS2*) were observed at high relative abundance (up to 30%) in many of our samples, particularly in 2015 where the average relative abundance for koalas eating *E. viminalis* was 15%. Such high relative abundances are unprecedented in gastrointestinal microbiomes. *Ley et al. (2008a)* reported an average relative abundance of 1% across mammals generally, although higher relative abundances have been reported e.g., 5% from a domestic cow (*Bos taurus*) and a capybara (*Hydrochoerus hydrochaeris*, (*Ley et al., 2008a*), 6.7% and 9% for individual koalas (*Soo et al., 2014*; *Shiffman et al., 2017*), and 4.7% and 4.5% for domestic rabbits and American pikas (*Oryctolagus cuniculus*, *Zeng et al., 2015*); *Ochotona princeps*, (*Kohl et al., 2017*). Non-photosynthetic Cyanobacteria, now placed within the candidate phylum Melainabacteria (*Di Rienzi et al., 2013*; *Soo et al., 2014*), have only recently been recognized in gastrointestinal microbiomes, they are obligate anaerobic fermenters and syntrophic hydrogen producers which may benefit the host by the synthesis of vitamins B and K (*Di Rienzi et al., 2013*), and which are enriched in Kyoto Encyclopedia of Genes and Genomes (KEGG) Ortholog groups specific to amino acid metabolism, relative to other cyanobacteria (*Harel et al., 2015*).

The relatively low relative abundance of Proteobacteria (1%) in koala faeces from Cape Otway is less than that reported from faecal samples across mammals and herbivores generally (8.8% and 5%, *Ley et al., 2008a*; *Nelson, Rogers & Brown, 2013*) and lower than detected in the folivorous two-and three-toed sloths (*Choloepus hoffmanni* and *Bradypus variegatus*, 60%; *Dill-McFarland et al., 2016*. It is also substantially lower than that observed in zoo koalas by *Shiffman et al. (2017)* and *Barker et al. (2013)*) 15% and 2–9% respectively), but consistent with the observations of *Alfano et al. (2015)*.

Low-relative abundance microbes can also be functionally important in the gut microbiome (*Qin et al., 2010*). *Synergistes* is an anaerobic fermenter of some amino acids (*Allison, MacGregor & Stahl, 2015*), and by fermenting the toxic amino acid, mimosine, in the forage legume, *Leucaena leucocephala*, protects ruminant hosts from toxicosis (*Allison et al., 1992*). *Synergistes* is also a member of a consortium that can protect sheep from pyrrolizidine alkaloid toxicosis (*Lodge-Ivey et al., 2005*; *Rattray & Craig, 2007*) and can anaerobically degrade fluoracetate (*Davis et al., 2012*). *Shiffman et al. (2017)* suggested that *Synergistaceae* may play as-yet unknown roles in addition to those above in koalas, including

the degradation of plant toxins from *Eucalyptus*. On the basis of its metabolic potential and the relatively high abundance of *Synergistaceae* that they observed in koalas relative to most other gut ecosystems, they also identified it as the most likely core specialised member of the koala microbiota. However, this phylum occurred at lower concentrations in faecal microbiomes from Cape Otway, particularly in 2013 where mean abundances were as low as 0.0004% (*E. obliqua*). This suggests a key role of *Synergistaceae* in allowing the koala to subsist on *Eucalyptus*, as proposed by *Shiffman et al. (2017)*, is either not essential at least for some koala diets, or can be filled by other bacteria.

Another bacterial population previously identified by *Shiffman et al. (2017)* as discriminating the koala microbiota from the wombat microbiota is the family *S24-7* (phylum Bacteroidetes), which they observed at a mean relative abundance greater than 10% in zoo koalas, and which they linked to dietary specialisation in *Eucalyptus*. *Ormerod et al. (2016)* detected two trophic guilds ($\alpha$-glucan and plant glucan) among *S24-7* population genomes isolated from a koala; these genomes were remarkable for their large size relative to other *S24-7*. *Shiffman et al. (2017)* also detected a full suite of ureolysis genes in an *S24-7* genome accounting for ∼8% of the faecal microbial community. However, *S24-7* was not found amongst wild Cape Otway koalas, echoing the findings of *Barker et al. (2013)*, who detected S24-7 at only low abundance (0.07%) in the caeca of two wild koalas, but not in their faecal pellets.

In common with other culture-independent investigations of the koala microbiome (*Alfano et al., 2015*; *Barker et al., 2013*; *Shiffman et al., 2017*), our study also found that tannin-protein complex degrading bacteria (Enterobacteriaceae, Pasteurellaceae and Streptococcaceae) previously cultured by Osawa and colleagues (*Osawa, 1990*; *Osawa, 1992*; *Osawa, Blanshard & Ocallaghan, 1993*; *Osawa et al., 1995*) were rare and occurred at low abundance (4 and 1% for 2013 and 2015 collections respectively) in wild koala faeces.

In addition to community differences associated with diet, differences were also apparent between the two collection years. For instance, *Bacteroides* relative abundance was three × greater in *E. viminalis* communities in 2015 than in 2013, while *Parabacteroides* relative abundance was lower in *E. viminalis* but 47% greater in *E. obliqua* diets in 2015 compared to 2013. We speculate that these differences might be explained by differences in food availability, leaf type and/or nutritional status for *E. viminalis* koalas between 2013 and 2015. Therefore, observed differences in microbial relative abundance between collection years might be explained by a return to a "normal diet" after the peak of overbrowsing associated with peak koala population densities; this would be consistent with the diet composition analysis, which suggested that all koalas were eating *E. viminalis* to some extent in 2015. Periodic fasting through dietary restriction or seasonal hibernation has been found to alter the microbial community structure in mammals (*Clarke et al., 2012*). In particular, other studies have observed increases in acetate-producing bacteria including *Akkermansia muciniphila* during periods of fasting, and suggested that under these circumstances, host-derived polysaccharides such as mucins are used as a substrate to produce short-chain fatty acids that support the host (*Carey, Walters & Knight, 2013*; *Derrien et al., 2008*; *Sonoyama et al., 2009*). The relative abundance of *Akkermansia* was notably elevated (between 1 and 13%) in some of the koalas eating *E. viminalis* in 2013,

and this may indicate that these individuals were experiencing a shortage of food. However, despite the differences in microbial relative abundance associated with collection period, the gut microbial communities associated with the two different diets remained distinct throughout.

Here we have shown that even a seemingly subtle dietary change can modulate the microbiome of a specialist herbivore. Based on our results, we postulate that diet preferences and the availability of resources will substantially impact the structure of gut microbial communities of koalas more widely, with consequences for the efficiency of digestion of their complex diet. As strong regional differences have been observed in the diet composition of koalas across their geographic range, translocated and released koalas (e.g., after overpopulation, habitat degradation, or rehabilitation) and those treated with antibiotics may not have the same microbiome as natural populations. Thus, a priority area of research should be to determine whether the therapeutic or prophylactic alteration of koala microbiomes can assist management and welfare outcomes for koalas facing enforced dietary change. New conservation and management strategies could include the development of targeted inoculations, thereby facilitating an increased dietary breadth for koalas, as demonstrated by *Kohl et al. (2014d)* and *Kohl, Stengel & Dearing (2015)* in the specialist desert woodrat.

## CONCLUSION

We show that diet differences, such as a change in consumption from one to another congeneric tree species, can profoundly affect the gut microbiome of a specialist folivorous mammal, even amongst individuals within a single contiguous population.

## ACKNOWLEDGEMENTS

The authors would like to acknowledge and thank the following people: Desley Whisson, Scott Bevins, Jack Pascoe, Lizzie Corke, Shayne Neal and the Conservation Ecology Centre at Cape Otway, Manuel Delgado Baquerizo and Jasmine Grinyer. The technical staff at HIE for running samples including T-RFLP and dietary analysis including GC-MS and C:N analysis. The staff of the WSU NGS centre for following and optimising the V4 original Earth Microbiome protocol.

### Funding

This research was supported under the Australian Research Council's Linkage Projects funding scheme (project number LP140100751); an Australian Postgraduate Award to Kylie Brice; the Western Sydney Postgraduate top up award and a Paddy Pallin Science Grant Award from the Royal Zoological Society of NSW. The funders had no role in study design, data collection and analysis, decision to publish, or preparation of the manuscript.

## Grant Disclosures

The following grant information was disclosed by the authors:
Australian Research Council's Linkage Projects: LP140100751.
Australian Postgraduate Award.
Western Sydney Postgraduate top up award.
Paddy Pallin Science Grant Award.

## Competing Interests

The authors declare there are no competing interests.

## Author Contributions

- Kylie L. Brice conceived and designed the experiments, performed the experiments, analyzed the data, prepared figures and/or tables, authored or reviewed drafts of the paper, approved the final draft.
- Pankaj Trivedi conceived and designed the experiments, analyzed the data, contributed reagents/materials/analysis tools, prepared figures and/or tables, authored or reviewed drafts of the paper, approved the final draft.
- Thomas C. Jeffries and Christopher Mitchell analyzed the data, contributed reagents/materials/analysis tools, authored or reviewed drafts of the paper, approved the final draft.
- Michaela D.J. Blyton analyzed the data, authored or reviewed drafts of the paper, approved the final draft.
- Brajesh K. Singh contributed reagents/materials/analysis tools, authored or reviewed drafts of the paper, approved the final draft.
- Ben D. Moore conceived and designed the experiments, contributed reagents/materials/analysis tools, authored or reviewed drafts of the paper, approved the final draft.

## Data Availability

BioProject: PRJNA521666.

## Supplemental Information

Supplemental information for this article can be found online at http://dx.doi.org/10.7717/peerj.6534#supplemental-information.

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
