# Peer review of "The Koala (Phascolarctos cinereus) faecal microbiome differs with diet in a wild population"

_PeerJ, doi:10.7717/peerj.6534_

## Round 0.1 · original submission · Major Revisions

The reviewers have provided useful information, which I agree with, and that should be addressed prior to publication. Some new/complementary analysis is suggested, and some points deal only with a better explanation on how the work was conducted.

However, I also want to point out that your paper specifically makes a claim to: "...predict functions within the microbial community based on 80 previously identified Parabacteroides and Ruminococcaceae genomes."

I find this analysis could be very interesting, but your conclusions overreach what was actually done: a simple comparison between GH present in two bacterial groups, based solely on published genomes.

A much better inference of these functional roles could be made if you at least provide support for the relationship between your koala gut microbiome 16S rRNA gene sequences, and those from the sequenced strains used in your analysis. Are they similar or not? By what criteria? How much can we extrapolate from those genomes in the database to the actual bacteria living in the koala gut?

Reviewer 1 ·

Basic reporting

No comment

Experimental design

What was the spatial separation between the individuals feeding on these two diets? Is it possible that other factors other than diet (climate, microbial availability, etc.) could have contributed to these differences? It is known that spatial microbial dispersal can drive diversity differences: http://www.pnas.org/content/114/52/13768

For alpha diversity, please include collection as a main variable, not as a random effect.

Lines 283 – 292 - another way you could investigate the magnitude of differences between years would be to compare pairwise distance measurements between the two groups. One might expect pairwise distances between groups to be larger in 2013 than in 2015.

Validity of the findings

No comment

Additional comments

Line 95, 129, anywhere else throughout the paper – please make sure to always use “16S rRNA”, not just 16S.
Line 142, 460, anywhere else throughout the paper - the word “This” should typically have a noun after it to make sure it is clear what it refers to.
Line 153 – maybe divide this into two sentences, or change to “so these individuals”
Line 192 – what does this 4% refer to? 4% of total sequences?
Line 197 – change to “abundances”
Line 199 – change to “levels”
Line 242 – seems like a word is missing here. Using an in vitro digestion.. model?
Line 463 – change

Reviewer 2 ·

Basic reporting

This study contributes to the knowledge about Koala microbiome and its interaction with their diet. Professional English language used throughout.
Some considerations and corrections to the authors:
1)ABSTRACT
1.1 Following Peer J Standards the Abstract section should have Headings structured to be bold and followed by a period. Background. Methods. Results. Discussion. This instruction was not followed and should be corrected.
1.2 The abstract contains only 221 words and could be enriched with results and discussion about beta diversity analysis.
1.3 Are n=32 the faecal bacterial communities or the koalas?
1.4 The term foliage used in the manuscript could be used in the abstract to better describe what part of the food species is consumed.
1.5 Is the dietary pattern of the Koalas a question of “strongly preferred” or availability of the species in the place they live?Describe it better .
1.6 Indicate Permanova tests and results in the Figures 1, 2 and 3 and also Supplemental figures.
2. Introduction
2.1 The Background about Cape Otway and the Koalas provided a very good context to the article, but the information about Eucalyptus leaves chemical composition could be enriched with data from antimicrobial properties of essential oils of Eucalyptus species .
2.2 At line 117, please clarify "they" .

Experimental design

1.The nutritional quality evaluation of the foliage samples was very simple. No anti nutritional parameters were evaluated. Explain why.
2.The leaf collection method indicated that Epicormic and adult E. viminalis (n=16) and E. obliqua (n=11) leaves were collected at Cape Otway in September 2013. Was there no collection in 2015? How was food species consumption attributed in 2015? This can impact Diet Composition results for 2015.
3.Explain the rationale of using two methodologies (terminal restriction fragment length polymorphism, and 16S amplicon sequencing) and two different beta diversity analysis (Unifrac for 16S and Bray-Curtis for RFLP) for comparison.
4. Metadata supplied indicates that sex and age characterization of the Koalas are available. Were microbiome data analysed under these perspectives? Was Permanova performed for theses data?Indicate the results found
5. A supplemental map with detailed GPS location of the koala and the forest patches could be added to.

Validity of the findings

Figure 1 B is missing E obliquoa data. Figure 1A has 13 red dots and Figure 1B only 8. Please correct it.
About Figure 5. 2013 data is complete but 2015 is showing 15 not 18 koalas. Is there a data missing or plot mistake? Is there difference between E. obliquoa diet consumption in 2013 and 2015? Please show the results. This could indicate 3 different diets groups: E. vitamalis, E. obliquoa and Mixed one and could explain differently Figure 1 and 3. Perform Permanova with 3 diets groups.
Figure S3 shows Fungi and Archea data.Results are explained in the Supplemental materials, but it was not possible to find Captions of Figure 3 A and B. It is not clear the year of the data showed.
Lines 312 and 313 indicate results about genus otu levels but indicate figures 3A and B, which have Archea and Fungi data . Please correct it.
Figure S4 is dark and is difficult to visualize data. Please correct it and re upload it.
Review reference indicated at line 379 (Turnbaugh and Gordon 2009b) , as it does not mention the words carnivorous or herbivorous, they talk about lean and obese.
In the discussion authors state "a seemingly subtle dietary change" but the nutritional composition of the two diets are very different. Re - evaluate

Statements at lines 504-510 have no basement in the studied population or data shown. It is a speculation, it would be better in Discussion.

Additional comments

No comment

---

## Round 0.2 · Minor Revisions

Dear authors,
The manuscript is much improved. However, one reviewer has very minor requests. I look forward to your corrected version addressing those 3 minor points.

Best regards,

Christian Hoffmann

[Reviewer 1 ·

Basic reporting

No comment

Experimental design

No comment

Validity of the findings

No comment

Additional comments

The authors have done a good job addressing my comments. I have no other issues with the paper

Reviewer 2 ·

Basic reporting

The author answered well all the questions and performed the necessary alterations.
I would like to mention three small notes:
1)The reference David et al 2014 is referred as David et al 2014 a. I see no need of “a” as there is only one reference from David et al in 2014.
2) Figure S1 and S2 made the places and sampling clearer. However when comparing the identifications placed in the maps and the Koalas identification codes at table S2a it is possible to note absences. The 2015 map is almost complete (17 of 18 koalas), but the 2013 map only shows 6 Koalas. Where is the location of the other 8 koalas, are they together with the other?
3) I am curious about the identification of 4 koalas from 2013. They do not follow codes but “names” (Alf, Beast, Bella, Bilko). Why were they named? Are they wild or not? I think it would be better code them with letters than names

Experimental design

The author answered well all the questions and performed the necessary alterations.

Validity of the findings

The author answered well all the questions and performed the necessary alterations.

Additional comments

I am curious about the identification of 4 koalas from 2013. They do not follow codes but “names” (Alf, Beast, Bella, Bilko). Why were they named? Are they wild or not?

---

## Round 0.3 · accepted · Accept

Dear Kylie et al,

Congratulations!

As your manuscript is being accepted, please add a sentence to the end of the session “Analysis of bacterial community through rRNA gene sequencing and analysis” indicating where the sequence data was deposited, and the respective accession numbers/id.
Please pay particular attention to make sure the deposited data can be connected to the samples described in this study (sample IDs and/or indexes used and/or filenames etc). Perhaps a good place to connect this information would be adding it to supplemental table S2a.

Also, for thoroughness, add the information regarding the Koala K13 that doesn't have the GPS coordinates. An appropriate location would be line 166/167 or perhaps the legend of the corresponding Sup figure.
Perhaps something like (adapted from your rebuttal letter):
"The GPS coordinates for the koala identified as K13 from the 2015 collection were not recorded, and therefore it is not shown on the map. Its faeces were sampled in the same area as the other E. viminalis koalas."

Best,
Chris

#